# Sex Determination in Two Species of Anuran Amphibians by Magnetic Resonance Imaging and Ultrasound Techniques

**DOI:** 10.3390/ani10112142

**Published:** 2020-11-18

**Authors:** María José Ruiz-Fernández, Sara Jiménez, Encarnación Fernández-Valle, M. Isabel García-Real, David Castejón, Nerea Moreno, María Ardiaca, Andrés Montesinos, Salvador Ariza, Juncal González-Soriano

**Affiliations:** 1Departamento de Medicina y Cirugía, Facultad de Veterinaria, Universidad Complutense, Avenida Puerta de Hierro s/n, 28040 Madrid, Spain; mariajoseruizfernandez@gmail.com (M.J.R.-F.); isagreal@vet.ucm.es (M.I.G.-R.); 2Departamento de Biología Celular, Facultad de Biología, Universidad Complutense, Avenida José Antonio Novais 12, 28040 Madrid, Spain; sajime01@ucm.es (S.J.); nerea@bio.ucm.es (N.M.); 3Unidad de RMN—CAI Bioimagen Complutense, Universidad Complutense, Paseo de Juan XXIII 1, 28040 Madrid, Spain; evalle@ucm.es (E.F.-V.); dcastejon@ucm.es (D.C.); 4Centro Veterinario Los Sauces, Calle de Santa Engracia 63, 28010 Madrid, Spain; ardiaca.m@outlook.com (M.A.); amontesinosbarcelo@gmail.com (A.M.); salvadorariza@msn.com (S.A.); 5Departamento de Anatomía y Embriología, Sección Departamental de Anatomía y Embriología (Veterinaria), Facultad de Veterinaria, Universidad Complutense, Avenida Puerta de Hierro s/n, 28040 Madrid, Spain

**Keywords:** amphibians, anurans, sex determination, magnetic resonance, ultrasonography

## Abstract

**Simple Summary:**

Amphibians are of critical importance among vertebrates. They play a critical role in their ecosystems, are commonly used as environmental health indicators and are also considered as exotic pets throughout the world. Among amphibians, many anuran species are included in active conservation programs as they are listed as endangered species. Thus, it is important for veterinarians and biologists to examine their sanitary status and to find a non-invasive tool to evaluate the health status of these individuals, particularly the state of their reproductive system and to be able to carry out a sex determination in case of no sexual dimorphism. For the first time, we demonstrate that benchtop magnetic resonance imaging and high-resolution ultrasound are suitable non-invasive imaging techniques for an accurate sex determination of two anuran species. Both techniques allowed the identification of ovaries and testes. Therefore, our data constitute an important contribution for clinical diagnostic and conservation purposes in amphibians, as it is possible to distinguish males and females in a quick, safe and relatively easy way.

**Abstract:**

The objective of the present study was to evaluate whether gender determination in two amphibian species (*Kaloula pulchra* and *Xenopus laevis*) can be reliably carried out by means of magnetic resonance imaging (benchtop magnetic resonance imaging; BT-MRI) or ultrasound (high-resolution ultrasound; HR-US) techniques. Two species of healthy, sexually mature anurans have been used in the present study. Eight *Kaloula* (blind study) and six *Xenopus* were used as controls. Magnetic resonance imaging experiments were carried out on a low-field (1 Tesla) benchtop-MRI (BT-MRI) system. HR-US examination was performed with high-resolution equipment. Low-field BT-MRI images provided a clear and quantifiable identification of all the sexual organs present in both genders and species. The HR-US also allowed the identification of testes and ovaries in both species. Results indicate that BT-MRI allowed a very precise sex identification in both anuran species, although its use is limited by the cost of the equipment and the need for anesthesia. HR-US allowed an accurate identification of ovaries of both species whereas a precise identification of testes is limited by the ultrasonographer experience. The main advantages of this technique are the possibility of performing it without anesthesia and the higher availability of equipment in veterinary and zoo institutions.

## 1. Introduction

Amphibians are one of the most interesting groups among vertebrates in terms of the diversity of their reproductive modes. Globally, they include over 7000 species of frogs (*Anura*), 700 species of salamanders (*Caudata*) and 200 species of caecilians (*Gymnophiona*) [1]. More than 2000 amphibian species are listed as endangered [2,3,4]. Amphibians play a critical role in their ecosystems and are commonly used as environmental health indicators in their habitats worldwide [4,5,6]. They are also frequently kept as exotic pets throughout the world. To evaluate these species properly, veterinarians need to be able to perform a clinical evaluation and a proper diagnosis in case of pathological problems, particularly those of the reproductive system [7].

Many anurans, such as *Xenopus laevis* (commonly known as the clawed frog), display several forms of sexual dimorphism, including differences in size, skin coloration, texture, secondary sex characteristics (nuptial pads, vowel sac color spines, glands, etc.) and/or behaviors, allowing for an easy differentiation between males and females [8,9]. In African populations, the average size of adult *Xenopus laevis* females is a 110 mm snout to vent length (SVL) (maximum 130 mm SVL) and males are three-fourths as large; however, under laboratory conditions the range varied from 50 to over 140 mm. Other species, such as *Kaloula pulchra* (Banded bullfrog), show very little dimorphism, with males presenting calling behavior during the mating season and being slightly smaller than females, ranging in size from 54 to 70 mm SVL for males and 57–75 mm SVL for females. These differences, however, overlap and are difficult to assess, making this species almost monomorphic [10]. In this sense, the use of non-invasive approaches such as diagnostic imaging methods is of particularly great value for sex identification in the weakly dimorphic or monomorphic species. The startup of non-invasive methods for identification of biological sex in wildlife and experimental animals is important, among other things, for the demographic or genetic management of captive breeding colonies [4,5,6]. It is especially important for many endangered species that need attention. In this context of the current global decline and extinction of many species, one of the main strategies in the case of amphibians has been to establish captive breeding populations for reintroduction programs. To this end, the first step is the identification of the gender of the individuals present within the colony and this is not always easily accomplished, particularly when working with species that have monomorphic or weakly dimorphic secondary sexual characteristics [2,4,5,6].

Several articles describe the use of ultrasonography in amphibians for medical purposes [11,12,13], assessment of reproductive status and sex identification in frogs with better specificity for female identification [14,15,16,17]. Magnetic resonance imaging (MRI) has been previously used in frogs focusing on different functional aspects of the Central Nervous System (CNS) [18,19]. However, there are no previous reports on MRI studies for clinical evaluation or sex determination, or for the use of the benchtop MRI (BT-MRI) in these species.

The objective of the present study was to evaluate the accuracy of BT-MRI and high-resolution ultrasound (HR-US) and assess their applicability for sex differentiation in two anuran species, one with “well defined sexual dimorphism, and another with very limited differences” (*Xenopus laevis* and *Kaloula pulchra,* respectively). Particularly, we tried to determine if the reproductive organs of both sexes, male and female, can be reliably visualized by these two techniques and determine the technical parameters that would allow the best imaging of the different organs of interest.

## 2. Material and Methods

### 2.1. Animals

A total of 14 healthy anurans (good overall appearance, shiny skin without lacerations, damage or changes in the color and normal motility and standard reflexes), 8 sexually mature *Kaloula pulchra*, without previous identification of males and females (blind study) and 6 *Xenopus laevis* (three males and three females) (Figure 1) were used. *Xenopus laevis* were purchased from the European Xenopus Resource Centre (EXRC; EXRC@xenopusresource.org). *Kaloula pulchra* specimens were obtained from commercial pet suppliers. Animals were handled in accordance with the guidelines for animal research set out in the European Community Directive 2010/63/EU and following the recommendations of the European Commission for the protection of animals used for scientific purposes [20]. All procedures were approved by the local ethics committee.

### 2.2. Procedures—Imaging Techniques

#### 2.2.1. Magnetic Resonance Imaging

BT-MRI was performed at the Bioimaging Center of the Complutense, University of Madrid (UCM) using a 1-Tesla benchtop-MRI scanner (Icon (1T-MRI); Bruker BioSpin GmbH, Ettlingen, Germany). The BT-MRI spectrometer consists of a 1 T permanent magnet (without extra cooling required for the magnet) with a gradient coil that provides a gradient strength of 450 mT/m. The MRI system bed integrates both the oval-cylinder solenoid radiofrecuency (RF) coil (59 × 50 mm^2^) and the animal monitoring system, which allows an accurate animal positioning and a full control of vital signs. A previous physical evaluation was carried out to ensure that the animal was healthy and well hydrated. Hydration was assured during the procedure, as it is a critical point for amphibians. For the performance of BT-MRI studies of *Kaloula pulchra,* animals were anesthetized by intramuscular injection of alfalxalone (5 mg/kg, Alfaxan 10 mg/mL). In the case of *Xenopus laevis*, specimens were deeply anesthetized by immersion in 0.1% tricaine methanesulfonate solution (MS222, pH 7.4; Sigma-Aldrich, Steinheim, Germany). After anesthesia, animals were positioned inside the RF coil in sternal recumbence.

The BT-MRI experiment consisted of two-dimensional gradient-echo T1-weighted coronal images. Two-dimensional images were acquired using a Fast Low Angle Shot (FLASH) MRI-sequence, the most commonly used gradient-spoiled gradient-echo imaging sequence. The main selected parameters were: repetition time = 302 ms, echo time = 3.05 ms, pulse flip angle = 80°, number of averaged experiments = 6, field of view = 74 × 46 mm^2^, slice thickness = 1 mm and number of slices = 15. The acquired matrix size was 172 × 209, the reconstructed matrix size 370 × 230 (resolution 0.20 × 0.20 × 1.00 mm) and the total acquisition time ~5 min.

#### 2.2.2. Ultrasonography

HR-US examinations were performed with an Aplio i800 (Canon Medical Systems, Otawara, Japan) equipped with 2 linear transducers (PLT 1202 BT with 4.5–17 MHz frequency range and PLI 2002 BT with 8.8–22 MHz frequency range). In order to optimize the image quality of the smallest anatomical structures, the highest frequency was selected in all cases. The animals were handled and examined without anesthesia. Frogs were positioned first in dorsal recumbence to obtain HR-US images from a ventral access (“ventral acoustic window”), and later in ventral recumbence, to obtain images from a dorsal access (“dorsal acoustic window”). The transducer was placed directly over the animal’s skin, previously wetted with acoustic gel to improve ultrasound conduction. The coelomic cavity was examined in transverse and longitudinal planes, from cranial to caudal.

To avoid any subjective interpretation, all our images were analyzed by the same persons, which included personnel with long experience in diagnostic imaging and researchers with no previous training. Imaging interpretation was performed according to classical anuran anatomy descriptions [21].

## 3. Results

### 3.1. BT-MRI

A low-field (1T) BT-MRI system allowed to obtain non-invasive magnetic resonance (MR) images of the anuran coelomic cavity in a semi-automatic way, without manual optimization. Most of the structures (heart, lungs, kidneys, stomach, pancreas, spleen, small and large intestine) could be identified on the MR images in one viewing (see Appendix A, for further details)**.** The BT-MRI protocol developed in this study allowed us to obtain a suitable balance between a good signal-to-noise ratio, acceptable time per animal (~10 min), absence of motion artifacts (due to the use of fast imaging techniques) and good contrast of the main reproductive organs. In both species, the contrast obtained between tissues allowed an easy segmentation of several structures as testes and kidneys and therefore, an easy volumetric quantification (see Appendix A).

Figure 2A–F show the sexual organs of *Kaloula pulchra*. In both males and females, the gonads were identified in the dorsal part of the cavity, just cranial to the corresponding kidney. The female reproductive system mainly consists of ovaries and oviducts (Figure 2A–D). The pair of ovaries showed numerous follicles, which were visible on the ovary surface (Figure 2A,B). In the case of sexually matured females, it was possible to observe a great number of dark-colored follicles (Figure 2C,D). The oviducts were observed as two tubes, relatively long and coiled, that flow into the cloaca separately. The male reproductive apparatus consists of a pair of testes, *vasa efferentia* and two urinogenital ducts that open into the cloaca. Testes were easily visible and identified as two ovoid-elongated shaped structures, with a sufficient contrast in comparison to the surrounding tissues to be easily segmented (Figure 2E,F). See Appendix A for more detailed information on the reproductive system.

The *Xenopus* reproductive system identified by means of BT-MRI is shown in Figure 3A–F. In both males and females, the gonads were placed adjacent to the kidneys and were surrounded by consistent fat bodies (Figure 3). In the case of females, the identification was particularly easy, because of the recognizable image of the ovaries with well-developed follicles (Figure 3A–D). The oviducts were also easily identifiable. In males, two oval-shaped testes were placed on the dorsal aspect of the coelomic cavity (Figure 3E,F).

### 3.2. Ultrasonography

As pointed out before, all females included in our experiment had the ovaries full of follicles at the time of examination. Follicles appeared as a complex of anechoic (black) or hypoechoic (dark gray) rounded or oval areas separated by hyperechoic (bright) lines (Figure 4). Both structures were easily recognized not only by an experienced ultrasonographer, but also by researchers without previous training in the use of this imaging technique. Although the follicles were clearly identified in ventral and dorsal acoustic windows (Figure 5A,B), the ventral acoustic window was considered more adequate for female gonad examinations. On the contrary, the kidneys were better identified using the dorsal acoustic window (Figure 5B). The ultrasound examination was repeated in one female *Kaloula* just after oviposition (Figure 6). In this case, both ovaries with follicles were clearly identified as a bilateral hyperechoic structure with multiple hypoechoic foci.

Testes were identified in all males of both species (Figure 7A,B). Testes appeared as structures with an oval morphology, mid echogenicity (medium gray) and homogenous echotexture. Testes were located just ventral or ventrolateral with respect to the ipsilateral kidney. Kidneys had a similar appearance to testes, but their size was bigger, and the margins were slightly irregular, while testes had a smooth contour. A dorsal window offered an easier identification and differentiation of testes and kidneys than a ventral acoustic window. However, even when using very high-resolution equipment, as is the case of this study, the differentiation between both organs is difficult for non-experienced ultrasonographers.

## 4. Discussion

Due to the important role of amphibians in their ecosystems, as well as to their increasing importance as exotic pets, problems related to conservation and reproduction represent a big concern for veterinarians and biologists. Amphibian sex identification through different non-invasive techniques has been a subject of interest for many researchers. Genetic analysis is likely to be a difficult and expensive method due to the fact that most amphibians do not have distinct sex chromosomes and as oddities such as aneuploidy and polyploidy can occur [22]. Other sex identification methods in anurans, such as hormone measurement in blood, fecal or urine samples, present variable degrees of effectiveness, cost, technical availability and invasiveness [14,23,24,25,26,27,28,29]. In some cases, they are expensive, technically difficult, lightly discriminative, or even risky for the animal’s life or require sacrifice, making them unsuitable for conservation purposes. However, the use of imaging techniques for sex determination has been poorly investigated. In fact, there is only one previous study, focused on the use of Near Infrared Reflectance spectroscopic (NIRS) for sex identification in the Mississippi gopher frog (*Lithobates sevosa*). Results were successful in females with developing eggs, although males and undeveloped females remained indistinguishable [30]. For the first time, the present study investigated the applicability of two routine imaging techniques (BT-MRI and HR-US) for sex identification in two anuran species with a well-defined and a very limited sexual dimorphism (*Xenopus laevis* and *Kaloula pulchra*).

Magnetic resonance imaging (MRI) techniques have become more accessible and popular in recent years, for both veterinarians and conservation experts. In comparison with other imaging techniques, MRI constitutes a valuable tool that allows the easy visualization and differentiation of soft tissue structures. In addition, the possibility of an automatic self-adjustment of the equipment’s main technical parameters and the lower cost of the new low-field benchtop-MRI equipment has contributed to its more generalized use in preclinical research and in a daily veterinary clinic routine [31,32,33,34,35]. The present study is the first report of the use of BT-MRI for the purpose of sex determination. This technique allowed a fast and precise differentiation of the genital organs in the two species studied (Figure 2 and Figure 3), as well as the identification of most of the structures in the coelomic cavity. Because of its characteristics compared to other imaging techniques, BT-MRI would be ideal for monitoring reproductive organs in a wide variety of animals that do not have sexual dimorphism. We think that this approach could be a useful possibility in a high variety of species, including those with a low SVL range—20 mm or even lower. This opens an important opportunity for the captive breeding populations for reintroduction programs, since this will, for example, maintain adequate sex ratios in captive breeding programs that are sexing metamorphic individuals (which is not possible or very difficult using other techniques) and thus increase the efficiency of conservation programs. The BT-MRI dataset obtained allowed us to not only determine the sex of the studied specimens, but it also provided valuable additional information, such as the detection and identification of anomalies and pathologies if present, and volumetric information from sexual and non-sexual internal organs (see Appendix A). For example, it could be a very interesting and useful application in track cases of sex reversal in amphibians in response to pollution, something that it is threatening amphibian populations throughout the world. The use of BT-MRI required a minimal dose of anesthesia to keep the animal unconscious for 10–15 min. The need for anesthesia could be a limiting factor, especially when working with sensible species, but in the present study all the frogs woke up normally after a mild anesthesia. Additionally, anesthesia will be likely substituted in the future by simple sedation or restraint techniques such as those used with mice [36]. In any case, the advantages of BT-MRI clearly outweigh the possible limitations, as the results are clear, acute, of very good quality and require minimal training for their interpretation.

Ultrasonographic visualization of reproductive organs in anurans has been considered difficult, especially in males [13,14,15,16,17]. However, in our study, ultrasound examination allowed an easy identification of testes and ovaries in both anuran species (Figure 5, Figure 6 and Figure 7). Yet, the identification of testes required trained ultrasonographers to make a correct interpretation of the images. In our study, we used state-of-the-art HR-US equipment and two transducers with a frequency range up to 17 and 22 MHz, respectively, which provided excellent quality images from the skin to 3–5 cm deep, while the transducers previously used in similar studies had lower frequency ranges [14,16,17,37]. The high technical characteristics of the HR-US equipment and transducers used in this study can explain the discrepancies with previous studies on the feasibility of ultrasonography to visualize testes and ovaries in anurans, as pointed out in studies pertaining to reptiles [38]. In comparison with other imaging techniques, HR-US has clear advantages, such as the good differentiation of soft tissue structures, its relatively low cost and wide availability in veterinary and zoo institutions. In our experience, correct gentle animal handling was sufficient to carry out a complete HR-US examination of the coelomic cavity in a few minutes (between 1 and 5). Thus, the possibility of handling animals without anesthesia is another important advantage of the HR-US technique for sex determination in amphibians.

Another interesting issue to consider for ultrasound examination is the animal’s position, that can be in dorsal recumbence, using a ventral acoustic window [13,14,16] or in ventral recumbence, using a dorsal acoustic window [13]. We found that the ventral acoustic window was the most suitable to evaluate ovaries in females, while kidneys and testes were better examined using the dorsal acoustic window.

Limitations of ultrasonography are the presence of artifacts caused by gas, the effect of bone and other mineralized structures, which avoid the conduction of HR-US waves. This drawback was particularly evident in *Kaloula pulchra* as these frogs greatly inflate the lungs as a defensive mechanism, complicating the visualization of organs especially when using the ventral acoustic window. Gentle handling and being patient until the frog deflated after 10–30 s allowed the ultrasonographic examination to continue with minimal delay.

## 5. Conclusions

In summary, we show that both BT-MRI and HR-US are suitable non-invasive imaging techniques for the accurate sex determination in both *Kaloula pulchra* and *Xenopus laevis* species. BT-MRI required minimal training for image interpretation and could be applied to low SVL range animals, but compared to HR-US, has the drawbacks of the higher cost of the equipment and the need for anesthesia. HR-US also allowed an accurate identification of ovaries of both species, even with minimal training, but the precise imaging and identification of testes in males required experienced ultrasonographers. The main advantages of this technique, compared to BT-MRI, are the shorter time required to perform the study (a few minutes per animal), the possibility of not using anesthesia and the availability of appropriate equipment in many veterinary and zoo institutions. In addition, the interpretation of data depends on experienced technicians, and anesthesia is not mandatory. Thus, the use of one or the other depends on the objective in each case.

This study represents a promising initial step that could promote the widespread use of these non-invasive imaging techniques for sex identification in amphibians or the clinical diagnosis of reproductive problems of these species. Our data constitute an important contribution for clinical diagnosis and conservation purposes in amphibians. However, further studies in other anuran species are needed to confirm the suitability of these imaging techniques for sex determination in these individuals.

## Figures and Tables

**Figure 1 animals-10-02142-f001:**
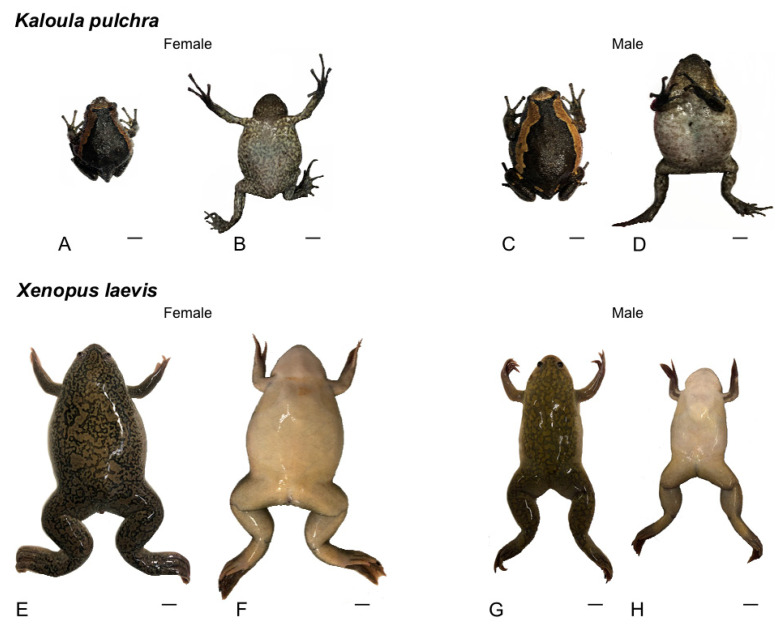
Photos of specimens in dorsal (left) and ventral (right) views of *Kaloula pulchra* (**A**–**D**) and *Xenopus laevis* (**E**–**H**) showing the differences between both species and females and males. Scale bars = 1 cm.

**Figure 2 animals-10-02142-f002:**
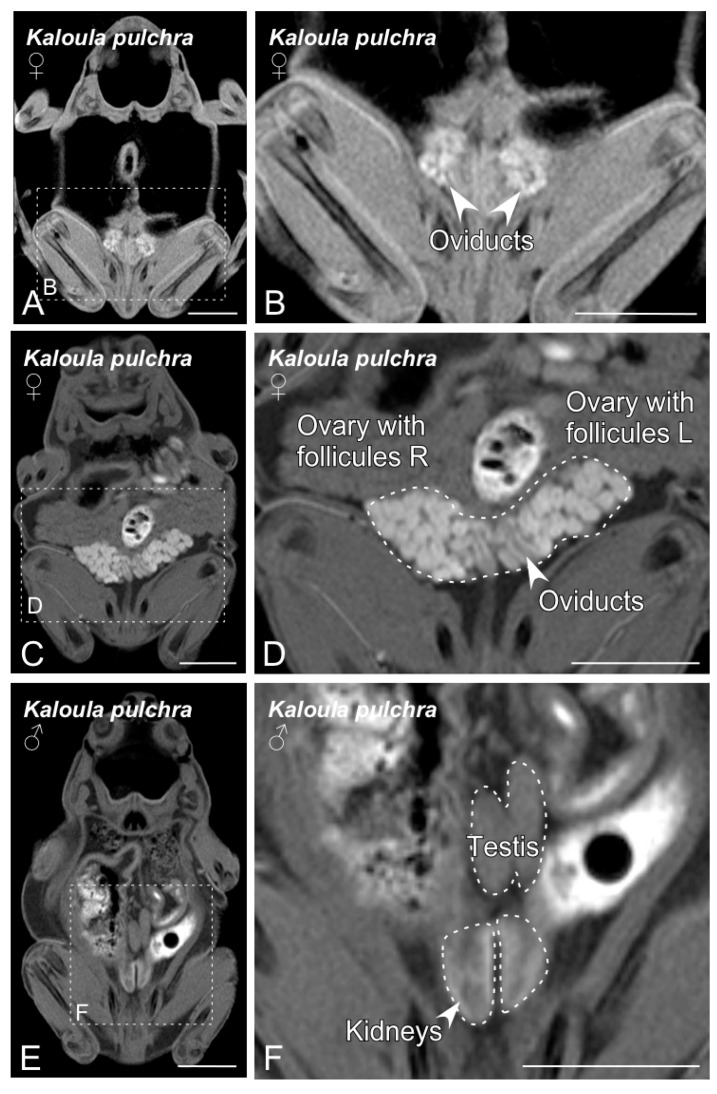
Magnetic resonance imaging (MRI) slices selected to classify *Kaloula pulchra* anurans by sex. Images from (**A**) to (**D**) show the main sexual structures identified in a non-gravid (**A**,**B**) and gravid (**C**,**D**) *Kaloula pulchra* female. Images (**E**) and (**F**) show testis and kidneys in a *Kaloula pulchra* male. The dashed lines in (**A**,**C**,**E**) indicate the magnification photos showed in (**B**,**D**,**F**). Scale bars = 1 cm.

**Figure 3 animals-10-02142-f003:**
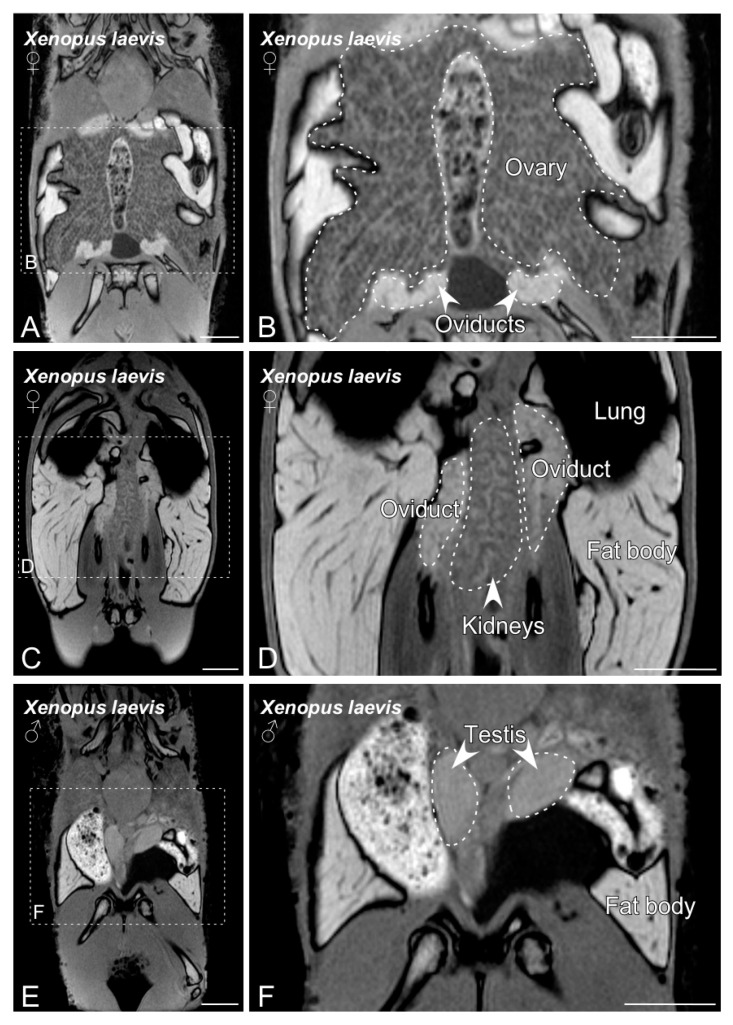
MRI slices selected to illustrate the main sex organs of the *Xenopus laevis* anurans. Images from (**A**) to (**D**) show the structures identified in a gravid *Xenopus laevis* female. Images (**E**,**F**) show testis and fat body in a *Xenopus laevis* male. The dashed lines in (**A**,**C**,**E**) indicate the magnification photos showed in (**B**,**D**,**F**). Scale bars = 1 cm.

**Figure 4 animals-10-02142-f004:**
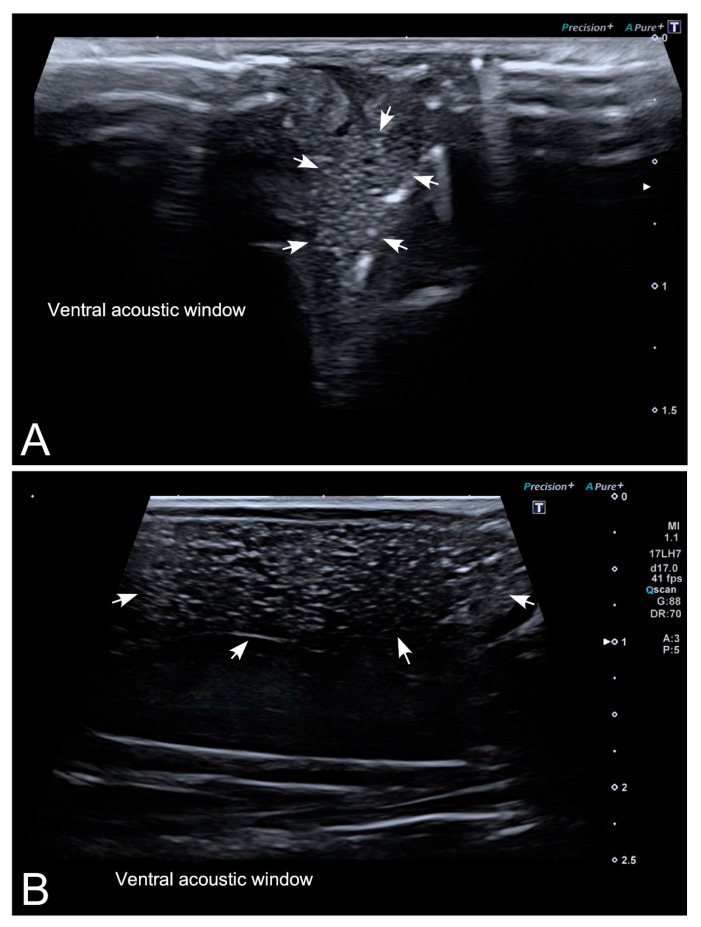
Representative high-resolution ultrasound (HR-US) images of a female *Kaloula pulchra* obtained in the transverse plane (**A**) and a female *Xenopus laevis* (**B**) obtained in the longitudinal plane using, in both cases, a ventral acoustic window. Follicles appeared as a complex of anechoic or hypoechoic rounded or oval areas separated by hyperechoic lines.

**Figure 5 animals-10-02142-f005:**
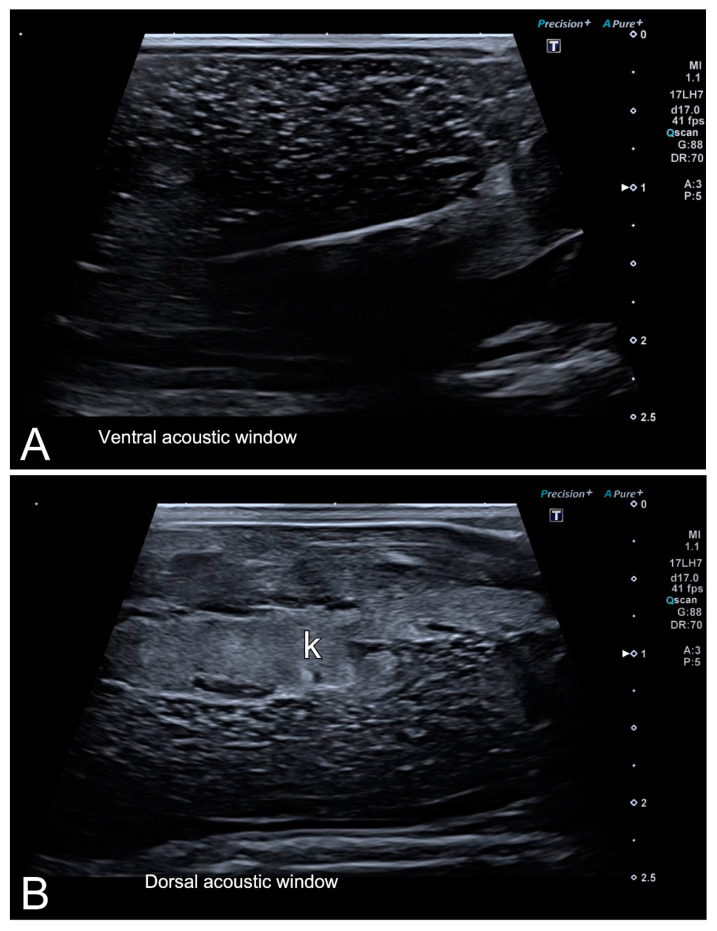
HR-US images of a female *Xenopus laevis* in the longitudinal plane using ventral (**A**) and dorsal (**B**) acoustic windows. Although the complex of follicles is clearly identified in both, the ventral acoustic window was considered more adequate for the examination of female gonads. Nevertheless, the kidney (k) was better identified using the dorsal acoustic window.

**Figure 6 animals-10-02142-f006:**
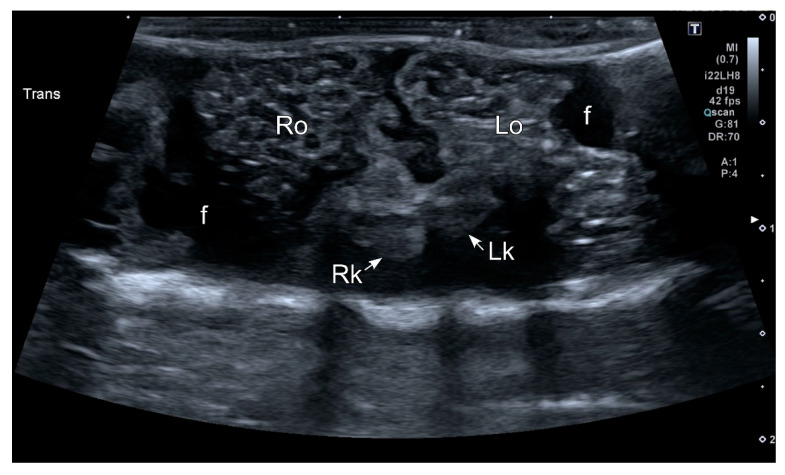
HR-US image of a female *Kaloula pulchra* obtained in the transverse plane using a ventral acoustic window. The image was taken just after oviposition. Both ovaries (right ovary: Ro; left ovary: Lo) appeared as hyperechoic structures with multiple hypoechoic foci which represent follicles. The kidneys are also visible (right kidney: Rk; left kidney: Lk). This animal had free fluid (f), which is considered normal in amphibians. The fluid around the ovaries improves its margin definition.

**Figure 7 animals-10-02142-f007:**
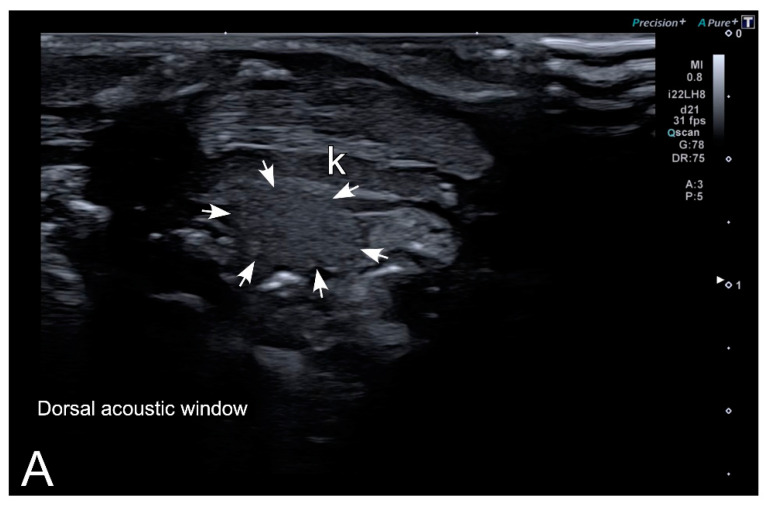
Representative HR-US images of *Kaloula pulchra* (**A**) and *Xenopus laevis* (**B**) males obtained in the longitudinal plane using a dorsal acoustic window. The testes (delimited by arrows) appeared as structures with oval morphology, mid echogenicity and homogenous echotexture. Testes were located just ventral or ventrolateral of the ipsilateral kidney (k).

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
