# Peer review of "Sex Determination in Two Species of Anuran Amphibians by Magnetic Resonance Imaging and Ultrasound Techniques"

_animals, 2020, doi:10.3390/ani10112142_

Round 1

Reviewer 1 Report

Ruiz-Fernández et al. present an interesting manuscript that investigates the use of BT-MRI and ultrasound for sexing amphibians. The authors were able to confidently interpret sex from the images generated from these techniques. The manuscript ends with highlighting the benefits of this technique for the care and management of amphibians throughout the world.

I believe the paper is in a good position for publication. I raise several points below that I would appreciate the authors address prior to publication. Essentially, my issues rely with how transferable this technique might be, given that the authors utilize two anurans (rather than a group that expands different natural histories and morphologies). I also think that this manuscript could benefit from mentioning some of the conservation issues associated with amphibians, and how sexing might help address them.

Line 53: You spend some time earlier in this paragraph discussing how amphibians are in decline. Why not mention how sexing can help in conservation too? For example: we usually do not know the sex of individuals until they reach sexual maturity. This can make maintaining adequate sex ratios in captive breeding programs difficult. Sexing individuals early (post metamorphosis?) could help increase efficiency of conservation programs.

Line 123: typo

Line 222: How transferable is your technique across anurans or amphibians, given that you restricted your study to two examples in a very diverse group of animals.

Line 224: Could your technique be used to track cases sex reversal in amphibians in response to pollution? I think this is another relevant topic for your paper, given that these issues also threaten amphibian populations throughout the world.

Reviewer 2 Report

As an amphibian taxonomist, I often struggle to identify the sexes of amphibians even when I evaluate the gonads of preserved specimens that lack sexual dimorphism under the microscope. Testing the application of new techniques to sex specimens is a fascinating and very useful idea.

The manuscript is very well written, structured, and supported with references and high-quality images. The results are well discussed. I marked a few corrections and placed a few comments on the PDF that I attach. Several images need to be slightly improved by changing the position of the labels, scales, and explaining the meaning of the dashed square in the legends.

I would discuss whether the size (SVL) of amphibians can impact the used techniques and how your used frogs recovered after the procedures. Did you lose any frogs?
